



# Cross-comparison of cloud liquid water path derived from observations by two space-borne and one ground-based instrument in Northern Europe

Vladimir S. Kostsov[1], Anke Kniffka[2], Martin Stengel[3], and Dmitry V. Ionov[1]

[1] Department of Atmospheric Physics, Faculty of Physics, St. Petersburg State University, Russia

[2] Institute of Meteorology and Climate Research, Karlsruhe Institute of Technology, Germany

[3] Satellite-based Climate Monitoring, Deutscher Wetterdienst, Offenbach, Germany

*Correspondence to: Vladimir S. Kostsov (v.kostsov@spbu.ru) and Anke Kniffka (anke.kniffka@kit.edu)*

**Abstract.** Cloud liquid water path (LWP) is one of the target atmospheric parameters retrieved remotely from ground-based and space-borne platforms using different observation methods and processing algorithms. Validation of LWP retrievals is a complicated task since a cloud cover is characterised by strong temporal and spatial variability while remote sensing methods have different temporal and spatial resolution. An attempt has been made to compare and analyse the collocated LWP data delivered by two satellite instruments SEVIRI and AVHRR together with the data derived from microwave observations by the ground-based radiometer RPG-HATPRO. The geographical region of interest is the vicinity of St.Petersburg, Russia, where the RPG-HATPRO radiometer is operating. The study is focused on two problems. The first one is the so-called scale difference problem which originates from dissimilar spatial resolutions of measurements. The second problem refers to the land-sea LWP gradient. The radiometric site is located 2.5 km from the coastline where the effects of the LWP gradient are pronounced. A good agreement of data obtained at the microwave radiometer location by all three instruments (HATPRO, SEVIRI and AVHRR) during warm and cold seasons is demonstrated (the largest correlation coefficient 0.93 was detected for HATPRO and AVHRR data sets). The analysis showed no bias of the SEVIRI results with respect to HATPRO data and a high bias (0.013-0.017 kg m$^{-2}$) of the AVHRR results for both warm and cold seasons. The analysis of LWP maps plotted on the basis of the SEVIRI and AVHRR measurements over land and water surfaces in the vicinity of St.Petersburg revealed the unexpectedly high LWP values delivered by AVHRR during cold season over the Neva river bay and over the Saimaa Lake and the abnormal land-sea LWP gradient in these areas. For the detailed evaluation of atmospheric state and ice cover in the considered geographical regions during the periods of ground-based and satellite measurements, reanalysis data were used. It is shown that the most probable reason for the observed artifacts in the AVHRR measurements over water/ice surfaces is the coarse resolution of the land-sea and snow/ice masks used by the AVHRR retrieval algorithm. The influence of a cloud field inhomogeneity on the agreement between the satellite and the ground-based data was studied. For this purpose, the simple estimate of the LWP temporal variability was used as a measure of the



spatial inhomogeneity. It has been demonstrated that both instruments are equally sensitive to the inhomogeneity of a cloud
field despite the fact that they have different spatial resolution.

**Keywords:** cloud liquid water path; remote sensing; ground-based microwave radiometer; RPG-HATPRO;
meteorological satellites; SEVIRI; AVHRR

## 1 Introduction

Cloud liquid water path (LWP) is one of the target atmospheric parameters retrieved remotely from ground-based and space-
borne platforms using different observation methods and processing algorithms. The ground-based LWP measurements by
microwave (MW) radiometers are de facto the reference data and the validation base for LWP measurements from space
since they have a precision that is superior to current satellite remote sensing techniques (Roebeling et al., 2008a). These
techniques are based on measurements from space either of the self-emitted microwave radiation or of the reflected solar
radiation in visible and near-infrared ranges. The MW radiation measurements deliver the information independently of solar
illumination conditions but only above water areas since the emissivity of land surface in the microwave region is highly
variable. The advantage of measurements of the reflected solar radiation is the capability to monitor the atmosphere over
water areas and land surface as well. The present study deals with the latter type of measurements carried out by two space-
borne instruments: SEVIRI (Spinning Enhanced Visible and Infrared Imager) and AVHRR (Advanced Very High
Resolution Radiometer).

The quantification of the accuracy of LWP retrievals from the observations by the satellite AVHRR instrument has
been done by Jolivet and Feijt (2005) who used ground-based microwave radiometer data as a reference. Recently, several
studies have been done that were focused on the comparison of the cloud liquid water path values derived from the space-
borne observations by the SEVIRI instrument and ground-based microwave radiometers operating at different locations in
Europe (Roebeling et al., 2008ab; Greuell and Roebeling 2009; Kostsov et al., 2018). All these studies demonstrated the
general agreement between the compared data and revealed the main problems relevant to the process of validation of LWP
values derived from satellite measurements. Validation of satellite LWP retrievals is a complex task since cloud cover is
characterised by strong temporal and spatial variability while remote sensing methods have different temporal and spatial
resolution. The pixel size of the satellite observations is of the order of several kilometres, but the information provided by
the ground-based microwave radiometer refers to an area of a horizontal size of a few dozen meters. This fact is the origin of
a so-called "scale difference" problem. In order to make the results of measurements suitable for comparisons, it is necessary
to perform time averaging of ground-based data over the interval that is approximately equal to the time of the cloud
movement across the satellite pixel area. Greuell and Roebeling (2009) have proposed to perform averaging of the ground-
based MW measurements with a Gaussian weight function, by using a time scale that is longer by a factor of 3–15 than the
time of the cloud movement across the validation area. However, the study by Kostsov et al. (2018) detected no influence of



the duration of averaging period on the results of comparison (20 minute and 1 hour periods were considered). There are several factors influencing the results of LWP comparisons which are coupled with the scale difference problem and therefore should be mentioned: cloud field inhomogeneity, multi layer clouds, uncertainty of the wind speed at a cloud top, and the spatial variations of the surface reflectance in case of optically thin clouds.

Along with the scale difference problem, there are a number of problems arising from measurement geometry if an instrument operates onboard geostationary satellite and the measurements in the Northern latitudes are considered:

- large viewing angles result in observation of a cloud from its side rather than from top;
- large viewing angles are the reason for the considerable parallax effect (the horizontal displacement of a cloud viewed by a ground-based radiometer in a satellite image);
- large solar zenith angles in winter cause the increase of the retrieval errors of the satellite methods based on measurements of reflected solar radiation.

It should be stressed that the specific instrumental and algorithmic error sources are beyond the scope of our consideration.

All enumerated problems and factors are well-known and have been previously analysed both qualitatively and quantitatively. However some of them require further investigation due to the large variety of observational conditions. For
example, the study by Kostsov et al. (2018) pointed at the complexity of the scale difference and parallax problem in the coastline area. It has been also indicated that a more extensive database is needed for comparisons of ground-based and satellite LWP observations at Northern latitudes, especially for analysis of the winter season in order to explain, in particular, the differences between the observational and reanalysis-based LWP diurnal cycles. Additionally, reasons for occasional very large discrepancies between the ground-based MW radiometer and SEVIRI data have still to be confirmed.

The present article is an extension of the study by Kostsov et al. (2018) in which a joint analysis of the LWP values obtained from observations by the SEVIRI satellite instrument and from ground-based observations by the RPG-HATPRO (Radiometer Physics GmbH – Humidity And Temperature PROfiler) microwave radiometer near St.Petersburg, Russia (60N, 30E) has been made. The present article is focused mainly on the scale difference problem and related factors. The collocated SEVIRI and RPG-HATPRO data sets are combined with the LWP measurements by the satellite instrument
AVHRR which has noticeably higher spatial resolution than the SEVIRI instrument. The cross-comparison of LWP values obtained from three different sources has been chosen as an appropriate tool for an expanded analysis of the consistency of ground-based and satellite data.

Previously, there were studies which included comparison of LWP values derived simultaneously from different platforms. Dong et al. (2002) presented the results of a measurement campaign aimed at the investigation of the low-level
stratus cloud microphysical properties observed by ground- and satellite-based remote sensors and aircraft in situ instruments. Space-borne radiance measurements by the eighth Geostationary Operational Environment Satellite (GOES),



the ground-based MW radiometer, cloud radar and ceilometer measurements and the air-borne cloud droplet spectra measurements have been considered. A total of 10 hours of simultaneous data from the three platforms have been analysed. LWP was one of the target parameters for comparison and the results derived from the aircraft observations were taken as a baseline. The mentioned study is a very good example of how multi-platform observations that have different spatial and temporal sampling can be combined, made consistent and compared.

Also, it is necessary to mention previous studies in which the LWP data from SEVIRI and AVHRR have been intercompared, in particular the paper by Roebeling et al. (2006) which determines if SEVIRI can be used together with AVHRR to build a consistent and accurate data set of cloud optical thickness and cloud liquid water path over Europe for climate research purposes. Roebeling et al. (2006) evaluated the effects of recalibration, spatial resolution, and viewing geometry differences on the SEVIRI and AVHRR cloud property retrievals. Several important conclusions have been made. First of all, it has been shown that LWP values derived from SEVIRI and AVHRR observations differ significantly when the operational calibrations provided by the satellite operators are used. By means of recalibration, these differences can be considerably reduced.  The differences in spatial resolution and viewing geometry have a much smaller effect on the comparability of SEVIRI and AVHRR retrievals. Also, it has been suggested that over north-western Europe the SEVIRI retrievals are more sensitive to errors due to unfavourable viewing conditions; first, because SEVIRI has a large viewing zenith angle over this region, and second, because the scattering angle is close to 180$^\circ$, i.e., backscatter direction, for about 10% of the observations. Since over north-western Europe the viewing zenith angles of SEVIRI are large, Roebeling et al. (2006) expected that especially for early morning, late afternoon, and winter observations the cloud property retrievals from SEVIRI would have a much larger uncertainty than those from AVHRR. All these findings are taken into account in the present study.

## 2 Dataset description

The detailed description of the RPG-HATPRO and the SEVIRI datasets that are used in the present study has been presented in the article by Kostsov et al. (2018). Here we briefly enumerate the most important points:

1) The time interval 1 December 2012 – 30 November 2014 was taken for the analysis. It was divided in two seasonal periods: "WH" (warm and humid) which included May, June, July, August, September, October, and "CD" (cold and dry) which included November, December, January, February, March and April.

2) Two geographical areas were considered: the so-called "large terrain" and "small terrain". The large terrain had the size of about 200x200 km$^2$ with the city St.Petersburg at its centre and comprised parts of the Gulf of Finland, Karelian Isthmus, Ladoga Lake and the region to the South and South-West of St.Petersburg. The small terrain was centred at the MW radiometer location and its size was about 20x20 km$^2$. The radiometer is located close to the shore of the Gulf of Finland at a distance of 2.5 km from the coastline.



3) The high quality ground-based MW measurements were taken as a main criterion in the selection procedure of the collocated data (rain-free days only, no gaps in observations, and the successful convergence of the iterations of the retrieval process for every single measurement).

4) Simultaneously with synchronisation between the HATPRO and the SEVIRI values of LWP, the control of the cloud phase was made on the basis of the cloud parameters delivered by SEVIRI: only clear sky cases and only liquid phase clouds were selected for the analysis.

5) The total number of days of the collocated measurements was 210, including 120 days for the WH season and 90 days for the CD season.

6) The sampling interval of the ground-based measurements was taken as 10 s. The time averaging of the ground-based values was made over two intervals: 20 min and 60 min. The corresponding data sets were designated as $HAT_{10-20}$ and $HAT_{10-60}$.

In the present study, along with the data sets used previously, the LWP data delivered by the satellite instrument AVHRR are analysed. These data have been extracted from the Cloud_cci AVHRR-PM data set which is described by Stengel et al. (2017a). A scientific description of the data is given in the paper by Stengel et al. (2017b). The characteristics of the data subset extracted for the present study are the following:

1) The data version is the official V3, the access date is February 2019.

2) The geographical region has been selected as 3°×3° box centred at the MW radiometer location point (59.88107°N, 29.82597°E).

3) The subset contains sampled data on a regular grid (no averaging done). In each grid cell, the AVHRR pixel with the smallest satellite zenith angle is collected.

4) The AVHRR data are based on AVHRR GAC (Global Area Coverage) resolution with a footprint size of 1x4 km and a sampling distance of about 4 km.

Several important notes relevant to the selection of the AVHRR data which match the HATPRO and SEVIRI measurements should be made. We collected only the cases with liquid phase clouds (the cloud phase parameter cph=1) and the clear-sky cases (cph=0). For cph=0 all LWPs were assigned zero values. The additional criterion of the data selection was based on the analysis of the cloud detection uncertainty (CDU) described by the cloud mask uncertainty parameter cmask_asc_unc ("asc" indicates the ascending mode of measurements). All measurements with cmask_asc_unc greater than 30% were excluded from consideration. In the study by Keller et al. (2018), the value of 35% was taken for the cloud detection uncertainty limit. As stated by Keller et al. (2018), the value of 35% was "somewhat arbitrary but mainly based on analysing the relative frequency of cloud detection uncertainty which yielded a bimodal distribution when including all cloudy pixels, with 35% being approximately the value separating the more certain from the more uncertain clouds". In the present study, we also obtained bimodal distribution of cloud detection uncertainty for cloudy pixels (cph=1) corresponding





to land surface. In our case, 30% seemed to be a good approximation of the value separating the more certain from the more uncertain clouds. For the cloudy pixels corresponding to water area, the distributions of the cloud detection uncertainty were not bimodal. However the value of 30% looked reasonable for these pixels also. In order to be consistent, we applied the selection criterion based on the CDU analysis to the clear-sky pixels (cph=0) also with the value of 30% as a threshold for filtering out uncertain measurements.

The location of the ground pixels of the SEVIRI and AVHRR observations is shown in Figs. 1 and 2 for the large and small terrains. In the large terrain, there are four relatively large water areas that are covered by the AVHRR pixel grid: parts of Ladoga Lake in the North-East and of Saimaa Lake in the North-West, the Neva river bay in the centre and part of the Gulf of Finland in the West. The SEVIRI grid covers the Neva river bay in the centre and part of Ladoga Lake in the North-East. One can notice that the density of the AVHRR grid is higher than of the SEVIRI grid. The small terrain contains 9

SEVIRI pixels and 12 AVHRR pixels. The spacing of the SEVIRI grid is about 7 km, and the spacing of the AVHRR grid is about 4 km. The pixels with numbers 243 (SEVIRI) and 1861 (AVHRR) are the nearest to the radiometer site. It should be noted that all comparisons have been made for the ground pixels, which are nearest to the radiometer site. If other pixels are considered, they will be mentioned explicitly. The SEVIRI observations are made every 15 min under sun illumination conditions. The AVHRR observations over St.Petersburg are made twice per day but only once under sun illumination

conditions at about local noon (10-11 h UTC).

The spatial resolution of the AVHRR measurements is higher than that of the SEVIRI measurements, ≈ 1 km instead of 7 km in this location. Therefore in order to correctly compare the LWP values derived from the HATPRO observations with the AVHRR data, the ground-based data should be time-averaged over the interval that is shorter than the interval used for the comparison with the LWP values obtained from the observations by SEVIRI. We have chosen two time intervals

equal to 5 min and 10 min. So, all in all, the present study uses four HATPRO data sets with different averaging intervals (5, 10, 20 and 60 min). The sampling interval of the initial HATPRO data is equal to 10 s, that is why the data sets are designated as $HAT_{10-5}$, $HAT_{10-10}$, $HAT_{10-20}$, and $HAT_{10-60}$. Since the only one sampling interval of the initial data was considered, below we shall omit its indication and keep only the indication of the averaging interval. The example of the HATPRO data flow is presented in Fig. 3 in the form of running average values corresponding to different averaging

intervals. The selected time slot of 2.5 h on 2 July 2014 contains 5 instantaneous measurements by SEVIRI and 1 measurement by AVHRR. First of all, it should be noted that the LWP obtained by the radiometer is highly variable: for 5 min averaging interval, the LWP range is 0-0.4 kg m$^{-2}$. The second important note is that none of the values of the averaging interval can be given an evident preference from the point of better agreement with the satellite data. We also pay attention to the fact that there can be gaps in SEVIRI data set if certain LWP values are rejected due to selection criteria (if clouds are

not purely liquid, for example). As one can see in Fig. 3, the SEVIRI measurements at about 578.46 and 578.47 fractional day are absent. The panel "b" of Fig. 3 illustrates the position of collocated measurements of all three instruments on the





time axis. Since this study utilises the data sets prepared previously (Kostsov et al., 2018), the HATPRO selected and averaged data are primarily synchronised with the SEVIRI data. So, the time mismatch with the AVHRR data is larger but normally do not exceed 15 min. The HATPRO selected and averaged data are marked by crosses in Fig. 3b. One can see that the variability of $HAT_5$, $HAT_{10}$ and $HAT_{20}$ data is rather large even on the 15 min time scale. The $HAT_{60}$ values are also far from constant, however their range is much smaller (0.030-0.065 kg m$^{-2}$) and the SEVIRI and the AVHRR LWP values fit into this range. Concluding this section we would like to emphasise once again the importance of the scale difference problem which has been illustrated by Fig. 3.

## 3 Land-sea LWP gradient

The inhomogeneity of the cloud field at scales of a ground pixel of a satellite instrument is one of the considerable sources of discrepancy between the ground-based and satellite data. This inhomogeneity can have a meteorological origin or can be caused by the interactions between the atmosphere and different types of the underlying surface. The HATPRO radiometer is located close to the coastline of the Gulf of Finland and in the previous study by Kostsov et al. (2018) the land–sea gradient of LWP was clearly revealed by the SEVIRI observations in the vicinity of the radiometer site: higher LWP over land, lower LWP over water; the magnitude of the land–sea difference for mean LWP values calculated for the 2-year period 2013-2014 was about 0.040 kg m$^{-2}$, which is about 50% relative to the mean value over land. It should be mentioned that the land–sea differences of cloud characteristics in Northern Europe were detected earlier by Karlsson (2003) who compiled regional cloud climatologies covering the Scandinavian region on the basis of processing data from the AVHRR instrument for the period 1991–2000. During the spring and summer seasons, as a contrast to winter and autumn conditions, much less cloudiness was found over seawater and major lakes. It was suggested that the cold sea surface temperatures in the Baltic Sea (especially in spring and early summer due to inflow of cold fresh water from melting snow) lead to a considerable stabilization of near-surface layer of the troposphere. This explanation agrees well with what was detected for the St.Petersburg region in the study by Kostsov et al. (2018): the land–sea gradient in the mean LWP values for the cold and dry season was noticeably lower than for the warm and humid season.

In order to check whether this effect is present in the AVHRR observations in the vicinity of St.Petersburg, we plotted the maps of mean LWP values obtained by AVHRR for the small terrain (12 pixels) for three scenarios: the whole 2-year period, the WH season and the CD season, see Fig. 4. It should be noted that we used all available measurements when liquid water clouds or clear cases were detected regardless of the synchronisation with the HATPRO selected measurements. It means that rain events with high LWP might have been also included. The calculations for each pixel were made independently; therefore the number of averaged values per AVHRR pixel slightly varies: 248-305 for the 2-year period, 166-224 for the WH season and 70-90 for the CD season. The average time of AVHRR measurement over St.Petersburg is 0.454 in terms of the day fraction (10 h 53 min UTC) with the standard deviation of 0.021 which is about 30 min. For



plotting the SEVIRI LWP maps, in order to keep consistency between the spatial distributions obtained from the two satellite instruments, we selected one LWP value per day from SEVIRI observations which was measured at a time of 0.458 in terms of the day fraction. The number of averaged values for SEVIRI pixels also slightly varies: 487-509 for the 2-year period, 313-333 for the WH season and 166-179 for the CD season. The spatial distributions derived from these data are also plotted in Fig. 4.

Fig. 4 clearly demonstrates that the LWP spatial distributions over the small terrain obtained by the two satellite instruments are very similar for the WH season, but noticeably differ for the whole 2-year data set, and considerably differ for the CD season. The most important fact is the opposite direction of the LWP gradients for the CD season: while the AVHRR observations revealed the general increase of LWP from south-west to north-east, the SEVIRI observations show a decrease of LWP in this direction. It means that the AVHRR measurements for these time period show an opposite effect to the one described by Karlsson (2003) and Kostsov et al. (2018): the LWP amount derived by AVHRR over water area is higher than over land. It is important to emphasise that we do not analyse the detailed structure of the mean LWP maps since the number of initial data is not large. The gradient demonstrated by the AVHRR observations is the most striking for the CD season: the lowest LWP value over land is about 0.050 kg m$^{-2}$ and the highest LWP value over water reaches 0.170 kg m$^{-2}$. In contrast, the mean LWP values obtained from the SEVIRI observations are within much smaller range of 0-0.06 kg m$^{-2}$ for the CD season. For the WH season, the ranges of the mean LWP values obtained by the two satellite instruments are nearly the same (0.05-0.11 kg m$^{-2}$ for SEVIRI and 0.07-0.13 m$^{-2}$ for AVHRR) and the gradients are similar and demonstrate in general lower LWP values over water area (the North-Western part of the terrain) and higher LWP values over land (the South-Eastern part of the terrain). This behaviour is in accordance with the results of the studies by Karlsson (2003) and Kostsov et al. (2018). One important conclusion can be derived from the obtained maps of the mean LWP quantities: the comparison of data should be made absolutely separately for the WH and CD seasons, and the special attention should be paid to winter conditions when the differences between the SEVIRI and AVHRR data are the most pronounced. The reason for these differences is discussed below.

**4 Seasonal features at the radiometer location**

The number of synchronised HATPRO-SEVIRI-AVHRR measurements was 63 during the WH season, and 53 during the CD season. The main statistical characteristics relevant to the agreement of the data are given in Tables 1 and 2. The bias $b$, RMS difference $s$ and the bias-corrected RMS difference $s_0$ were calculated as follows:

$$b = \frac{1}{N}\sum_{j=1}^{N}\left(x_j - y_j\right)$$

(1)



$$s = \sqrt{\frac{1}{N} \sum_{j=1}^{N} (x_j - y_j)^2}$$

(2)

$$s_0 = \sqrt{\frac{1}{N} \sum_{j=1}^{N} (x_j - y_j - b)^2}$$

(3)

where $N$ is the number of data pairs, $x$ and $y$ are the compared quantities. First of all, there is a clear difference in the correlation coefficients for the satellite and the ground-based data for the WH and CD seasons: the AVHRR-HATPRO correlation coefficients for the warm season are higher than for the cold season, and for the SEVIRI-HATPRO datasets the situation is opposite. However for both seasons, the AVHRR-HATPRO data sets have the highest correlation coefficients reaching the value of 0.88-0.93. For the WH season, there is a clear dependence of the SEVIRI-HATPRO correlation coefficient and of the RMS difference from the averaging interval of HATPRO measurements: the longest averaging interval corresponds to the highest correlation coefficient and to the lowest RMS difference. At the same time, there is no influence of the averaging interval on the bias for the WH season. For the CD season, there is no influence of the averaging interval on any of the considered statistical characteristics for all datasets.

The bias of the SEVIRI data is nearly zero for both seasons of observations. The bias of the AVHRR data is considerable and it is larger for the CD season than for the WH season. The RMS differences between the satellite and the ground-based data (SEVIRI-HATPRO and AVHRR-HATPRO) are noticeably larger for the CD season than for the WH season. However these RMS differences are smaller than the RMS differences between the data provided by two satellite instruments (SEVIRI-AVHRR). Bias-corrected RMS differences are very close to the RMS differences for all cases. This is a kind of indication of the dominant character of the random component of the total discrepancy between the results obtained by all instruments.

One of possible explanations of the presented above results can be the following. During the warm and humid season, the convective clouds are much more frequent than during the cold and dry season and, as a result, the cloud field can be fragmented on the scale of several kilometres. In this case the size of a ground pixel of a satellite instrument plays an important role. If the pixel size is large and the fragmented cloud field is viewed from a satellite, one can imagine a situation when the ground-based radiometer located inside the pixel observes only cloud or only clear sky depending on wind direction. In this case the discrepancy between the satellite and ground-based data is expected to be very large. The smaller the pixel size is, the better agreement between the satellite and ground-based data is detected. Therefore, for the WH season, the agreement between the AVHRR data and the HATPRO data is much better than between the SEVIRI data and the HATPRO data. For the cold and dry season, when the clouds are predominantly stratiform, one should expect less influence of the pixel size of a satellite instrument on the agreement between the satellite and ground-based measurements. And we notice that the SEVIRI instrument with lower spatial resolution demonstrates higher correlation with the radiometer data



during the cold season than during the warm season.

In order to have the impression of the agreement of the ground-based and the satellite data during different months, we examine Fig. 5 where these data are shown as a function of day sequence number, which corresponds to the simple consecutive enumeration of days in the data sets. Also, the figure presents the distribution of days in the data sets over months: there are two sequences of consecutive months which correspond to 2013 and 2014 years. First of all, we note the

overall good agreement of all measurements during both seasons. The agreement for situations when LWP is zero or very close to zero (clear-sky cases) is almost perfect, the mismatches are very rare: for example, on day No 42 during the WH season, HATPRO and AVHRR showed clear case and SEVIRI did not; and on day No 39 during the CD season, the SEVIRI and HATRO detected clear sky but AVHRR did not.  However in both of these two cases the detected LWP values were low and constituted 0.033 kg m$^{-2}$. The bias of the AVHRR results for cloudy situations can be very well seen in the plots for both

seasons.

We have to remind that the high quality of ground-based observations was the basic criterion for selection of data for comparisons. The collocated data triplets (HATPRO-SEVIRI-AVHRR) were filtered out only in cases when the satellite observations reported the presence of ice clouds or mixed phase clouds. So, the evaluation of the data quality of space-borne measurements was not carried out except the analysis of the cloud detection uncertainty reported by AVHRR (the data with

CDU larger than 30% were removed). Now we analyse the LWP retrieval uncertainty which is one of the main quantities characterising the data quality and which is provided by both SEVIRI and AVHRR data processing algorithms. The LWP retrieval uncertainty (LWPU) is plotted as a function of the LWP value in Fig. 6 for the AVHRR and SEVIRI instruments and for different seasons. The general comparison of the distribution of data points shows the following main differences between the AVHRR and SEVIRI data:

-    for AVHRR, there is only one data point with the LWP less than 0.01 kg m$^{-2}$ while for SEVIRI there are many of them (the cases with LWP=0 are not taken into account);

-    the LWP uncertainties reported by AVHRR are much lower than reported by SEVIRI;

-    for SEVIRI, there are several measurements of low LWP which have the relative uncertainty higher than 100%,  and the number of such measurements is larger during the cold season while there is only one AVHRR measurement with

an uncertainty higher than 100%.

There are also common features in the distributions of the AVHRR and SEVIRI data. First, the dependence of LWPU on LWP on a logarithmic scale is very close to linear for both instruments. And second, the data points for the cold season are more scattered than for the warm season. The reason for that is the larger number of unfavorable observational conditions in winter.

In order to analyse how the data agree within the limits of their declared uncertainty, in Fig. 7 we plotted the histograms of the ratio of the absolute difference between the satellite and the ground-based data to the LWP uncertainty


reported by the satellite instruments:

$$R = \frac{\left| LWP_\mathrm{s} - LWP_\mathrm{g} \right|}{LWPU_\mathrm{s}} \qquad (4)$$

where "s" and "g" denote the satellite and the ground-based measurements correspondingly. It should be mentioned that the
clear sky cases were excluded from this analysis if these cases were detected by a satellite instrument. As a result, the
number of remaining data pairs was rather small: 18-20 for the CD season and 23-35 for the WH season. The distributions
demonstrate that all SEVIRI measurements agree with the HATPRO measurements within the limit of 3·LWPU. For
AVHRR, the distributions have longer "tails" and for the WH season the value of $R$ reaches 4-8 in some cases. This is a
kind of indication of the fact that in these cases the LWPU can be strongly underestimated by the AVHRR retrieval
algorithm. For both satellite instruments, the maximum of distributions corresponds to the interval from 0 to 1 and this
maximum is well pronounced. The majority of space-borne results match the ground-based measurements within the limit of
2·LWPU: 80-95% during the CD season and 66-73% during the WH season. The mean values of $R$ for the CD season for
both instruments constitute 0.93. For the WH season, the mean value of $R$ constitutes 2.8 and 1.3 for AVHRR and SEVIRI
correspondingly. Accounting for all these quantities, we can come to the conclusion that the declared uncertainties match the
differences between the satellite and the ground-based data during the cold season better than during the warm season. The
reason for such a result can be the difference in cloudy conditions in summer and in winter which was discussed above. Due
to the large probability of cumulus clouds, a cloud field can be fragmented on a scale of a ground pixel of a satellite
instrument. Fragmentation of a cloud field can be a source of additional discrepancy between the satellite and the ground-
based measurements.

## 5 Seasonal features over water areas

The most remarkable feature is the behaviour of the AVHRR results during the cold season over water area in the small
terrain as described in section 3, i.e. the unexpected land-sea gradient of LWP obtained by AVHRR and very high LWP
values over water area if compared to the SEVIRI measurements. In order to find out the reasons for this phenomenon, we
analysed the LWP maps for the large terrain as a first step. Fig. 8 shows the LWP maps based on the AVHRR observations
and plotted separately for the cold and warm seasons of 2013 and 2014. The attention is focused on four water areas which
are relatively large and are covered by the AVHRR pixel grid: the Neva river bay, parts of the Ladoga Lake, Gulf of Finland,
and Saimaa Lake. There is a striking difference between the maps corresponding to cold seasons of 2013 and 2014. For the
cold season of 2013, high LWP values were detected over three of four water areas mentioned above. The extremely high
LWP values can be seen over the Neva bay and over the Saimaa Lake. The LWP values over the Gulf of Finland are
considerably lower, but still noticeably exceed the LWP values over the surrounding land surface. And only the LWP values
over the Ladoga Lake are the same as the values over the neighbouring land surface. For the cold season of 2014, the LWP





over the Ladoga Lake and the Gulf of Finland are lower than for the land surface. And there is no noticeable difference between the LWP over land surface and LWP over the Saimaa Lake and the Neva bay. The LWP maps for the WH season of 2013 and 2014 shown in Fig. 8 are similar and demonstrate low LWP values over all four mentioned water areas if compared
to the land surface.

In order to understand the reasons for the land-sea LWP differences obtained by AVHRR during cold seasons of 2013 and 2014, we compare two months – March 2013 and March 2014. For characterisation of the weather conditions in the vicinity of St.Petersburg in 2013-2014, we use the weather reviews of the Russian North-West Administration on Hydrometeorology and Environmental Monitoring (http://www.meteo.nw.ru/articles/index.php?id=720, access date
29 November 2018). In March 2013, the weather conditions over the European part of Russia were determined mainly by anticyclones. As a result, March 2013 was abnormally cold on the territory of the European part of the Russian Federation and over the Central and Eastern Europe. Clear-sky conditions with large diurnal temperature ranges prevailed. There were only 3 overcast days. On average, the temperature was lower than normally in March by 2-7 K. The lowest air temperature in St.Petersburg was 256 K, the monthly mean temperature was 266 K. The temperature diurnal magnitude reached 15-25 K.
Total precipitation in St.Petersburg was 28% of the normal value. The weather in March 2014 was completely different from March 2013 and was determined mainly by cyclones, however during the first and the last week of the month the clear-sky conditions prevailed. The temperature was 267-273 K at night and 274-279 K in the day time. The mean monthly temperature in St.Petersburg was 275.5 K which is higher than the normal value by 4.4 K. In 2014, the spring season in St.Petersburg started 3-4 weeks earlier than normally. The differences in weather conditions in spring 2013 and 2014 are
illustrated by the ECMWF reanalysis data for the sea ice area fraction, see Fig. 9. The average monthly values of the sea ice fraction for March 2013 for two pixels (A and B) which refer to the Gulf of Finland were the highest in the winter season and constituted 0.7. For March 2014, these quantities were 0.014, which means that almost all ice had already melted. The situation for the Ladoga Lake (pixel C in Fig. 9) is different. In March 2013 and in March 2014, the ice fraction for pixel C was the same and constituted 0.96. It means that the Ladoga Lake can be considered as a kind of ice storage tank which was
not influenced by the early spring of 2014.

Taking into account different weather conditions during cold seasons of 2013 and 2014 we can suggest the following explanation of the AVHRR measurement results. The AVHRR retrieval algorithm uses a land-sea mask, and also it uses a sea-ice and snow mask. Due to low temperatures in March 2013, the snow and ice cover of the land and water areas was preserved. Probably, there were problems with the sea-ice and snow mask in areas 1 and 4, and the high reflectance of ice
and snow in combination with the large viewing angle of the AVHRR instrument resulted in erroneous high values of LWP during clear-sky-conditions. That is why AVHRR produced unexpected very high abnormal land-sea LWP gradient in the vicinity of the radiometer location in the cold season of 2013. The spring season in 2014 started early and the snow and ice cover disappeared quickly. Due to absence of the reflectance from ice and snow, the AVHRR measurements reported correct





values of LWP. We should note that during the CD season of 2013 there were no unexpected LWP results over the Ladoga lake and the abnormal land-sea LWP gradient for the Gulf of Finland was not very pronounced. Taking into account that the Neva bay and the Saimaa Lake are relatively small geographical objects while the Ladoga Lake and the Gulf of Finland have the scale of one hundred kilometers, we can suggest that the coarse resolution of the sea-ice mask can be a possible reason of the problem in areas 1 and 4.

The discussion of the problem of identification of clouds and ice/snow covered surfaces is beyond the scope of our study. We only note that it is an important problem which attracted much attention in the studies relevant to remote sensing of atmospheric state and composition from satellites. A very detailed overview on existing algorithms for cloud and snow detection on AVHRR imagery can be found in the paper by Musial et al. (2014).

It should be specially noted that we checked if the AVHRR data selection criterion based on the cloud detection uncertainty parameter influenced the results and conclusions which were obtained in the present study. No evidence of such

influence has been noticed. However, in order to be consistent with previous studies that used the AVHRR data selection criterion based on CDU (Keller et al., 2018) we did not discard the CDU control and rejected the AVHRR LWP results when CDU was larger than 30% as described in Section 2.

Concluding this section, we present Figs. 10 and 11 in which the LWP maps obtained from the AVHRR and SEVIRI retrievals are compared. The intersection of the pixel grids of the two instruments includes only two of the four investigated

water areas: the Neva bay and the Ladoga lake. In contrast to AVHRR, SEVIRI detected low LWP values over the Neva bay during the cold season of 2013 and of 2014 as well. So, the LWP land-sea negative gradient is clearly seen in 2014. It is not so pronounced in 2013 due to the fact that clear-sky conditions prevailed over the whole region. It should be noted that AVHRR reported considerably higher LWP values over land during the CD season of 2014. For the WH season of both 2013 and 2014, the differences between the AVHRR and the SEVIRI results are not so noticeable. Both satellite instruments

demonstrate negative LWP land-sea gradient and similar LWP values over the Ladoga lake. The LWP values over the Neva bay obtained by SEVIRI are lower than those obtained by AVHRR. The same situation exists for the land surface: AVHRR delivers higher LWP values everywhere.

## 6 Discussion of the scale difference problem

We already started the consideration of the scale difference problem when we analysed Fig. 3 and Tables 1 and 2. The

preliminary conclusion has been made that none of the values of the averaging interval for the ground-based measurements can be given an evident preference from the point of better agreement with the satellite data. However, if we consider the measurements during the WH season and the SEVIRI measurements only, the better correlation between ground-based and satellite data is detected for the averaging period of 1 h. This conclusion was obtained for the case when the HATPRO





measurements were synchronised with the SEVIRI measurements (the time gap between measurements was less than 2 minutes), but not with the AVHRR measurements. So, there was always a larger time mismatch between the HATPRO and AVHRR measurements which however did not exceed 15 minutes in all cases. The RMS value of the time mismatch was 9 min. Since the spatial resolution of the AVHRR measurements is higher than of the SEVIRI measurements, one can expect some influence of the time mismatch on the results of the data comparison. In order to check if this effect exists, we synchronised the HATPRO data with the AVHRR measurements and analysed the data agreement in this case. After synchronisation, the RMS value of the time mismatch decreased from 9 min to 83 s. The analysis has shown the absence of the expected effect: the AVHRR-HATPRO bias and RMS differences were close to the values indicated in Tables 1 and 3. The new values of the AVHRR-HATPRO correlation coefficients agree with the coefficients indicated in Tables 1 and 3 within the declared uncertainties. The same situation was true for the SEVIRI measurements, however one note should be made: when the HATPRO results were synchronised with the AVHRR results first, the correlation coefficients SEVIRI-HAT$_5$ and SEVIRI-HAT$_{10}$ changed considerably compared to the values in Tables 1 and 3. This is not surprising since time averaging over 5 and 10 minutes was done in order to study the agreement between the HATPRO and the AVHRR data rather than between the HATPRO and the SEVIRI data because of higher spatial resolution of the AVHRR measurements.

Since temporal averaging of the ground-based measurements is a necessary prerequisite for comparing them with the satellite data, the inhomogeneity of a cloud field and the uncertainty of a wind speed can be the reason for observed discrepancies. In order to investigate how these factors can affect the data agreement, in Fig. 12 we plotted the absolute difference between the ground-based and the satellite measurements of LWP as a function of the value of LWP variability estimate $V_e$, which was defined as follows:

$$V_e = \sum_{i \neq j} \left| HAT_i - HAT_j \right| \qquad (5)$$

where $HAT$ is the result of the LWP measurement by HATPRO, $i$ and $j$ indicate the averaging interval, in our case 5, 10, 20 and 60 min. It is evident that in the case of a homogeneous cloud field the $HAT$ values for different averaging intervals will be equal to each other, and $V_e$ will be equal to zero. The higher the $V_e$ value is, the stronger is the variability of a cloud field. Since there are several terms in the sum (5), the LWP variations of different temporal scales are accounted for. In our case the number of terms was 6:

$$V_e = \left| HAT_5 - HAT_{10} \right| + \left| HAT_5 - HAT_{20} \right| + \left| HAT_5 - HAT_{60} \right| + \\ \left| HAT_{10} - HAT_{20} \right| + \left| HAT_{10} - HAT_{60} \right| + \left| HAT_{20} - HAT_{60} \right| \qquad (6)$$

We analysed both the WH and CD seasons and obtained similar results, therefore only the results corresponding to the WH season are demonstrated and discussed. Each data point in Fig. 12 shows the LWP variability estimate at the moment of a measurement and the correspondent absolute difference between the satellite measurement of LWP and the



ground-based measurement averaged over 5, 10, 20 or 60 min. Obviously, one can not expect that the points will form any

definite functional dependence, since the variability estimate is not perfect, and, besides, there are many factors affecting the

data agreement other than inhomogeneity of a cloud field. Therefore, we can use Fig. 12 only for qualitative comparative

analysis. In Fig. 12 the logarithmic scale is used and for both satellite instruments we can notice the approximate linear

relation between logarithms of $D$ and of $V_e$. We calculated the correlation coefficients for $\ln(D)$ and $\ln(V_e)$ datasets and

obtained that for the WH season they are in the range 0.65-0.70 for the SEVIRI measurements and in the range 0.64-0.75 for

the AVHRR measurements. For the CD season, they are 0.50-0.65 for SEVIRI and 0.65-0.73 for AVHRR. Similarity of the

values can be an indication of the fact that for both considered instruments the results are equally sensitive to the

inhomogeneity of a cloud field. This conclusion is to a certain degree surprising since the SEVIRI measurements have lower

spatial resolution than the AVHRR measurements, so the results of LWP retrieval by SEVIRI were expected to be more

influenced by the inhomogeneity of a cloud field. However these results stay in agreement with the conclusions made by

Roebeling et al. (2006) who have shown that the differences in spatial resolution have a small effect on the comparability of

SEVIRI and AVHRR retrievals.

**7 Summary and conclusion**

The aim of the study was to compare and analyse the collocated cloud liquid water path (LWP) data provided by two

satellite instruments SEVIRI and AVHRR together with the data derived from microwave observations by the ground-based

radiometer RPG-HATPRO. The geographical region of interest is the vicinity of St.Petersburg, Russia, where the

RPG-HATPRO radiometer is operating. The radiometric site is located 2.5 km from the coastline of the Gulf of Finland

where the effects of the LWP horizontal gradient are pronounced. Two seasons were analysed: the warm and humid (WH,

May-October) and the cold and dry season (CD, November-April). Since the time averaging of the ground-based

measurements is a necessary prerequisite for the comparison procedure, four time intervals for averaging of the

RPG-HATPRO data were considered: 5, 10, 20, and 60 min. The number of synchronised HATPRO-SEVIRI-AVHRR

measurements was 63 during the WH season, and 53 during the CD season.

     The results of the comparison of the LWP values retrieved from the HATPRO, SEVIRI, and AVHRR observations

have shown the following:

1) The comparison of data should be made absolutely separately for the WH and CD seasons, and special attention should

be paid to winter conditions when there are considerable differences between the SEVIRI and AVHRR data obtained

     over several specific water areas.

2) The AVHRR and HATPRO data sets for the WH season have the highest correlation coefficients reaching the value of

     0.93. The overall good agreement of measurements by the three instruments is detected for both seasons; however the

     bias of the AVHRR data with respect to HATPRO was 0.013-0.017 kg m$^{-2}$ while the SEVIRI data had no bias. For the



WH season, the RMS differences SEVIRI-HATPRO and AVHRR-HATPRO were in the range 0.031-0.045 kg m$^{-2}$ and 0.035-0.037 kg m$^{-2}$ correspondingly. For the CD season, the RMS differences were larger than for the WH season, especially for AVHRR, and constituted 0.44-0.45 kg m$^{-2}$ for SEVIRI and 0.058-0.060 kg m$^{-2}$ for AVHRR.

3) The LWP uncertainties provided by the retrieval algorithms of both instruments match the differences between the satellite and the ground-based data during the cold season better than during the warm season. In some cases during the

WH season the LWP retrieval uncertainty is strongly underestimated by the AVHRR algorithm.

4) Both SEVIRI and AVHRR instruments demonstrated similar horizontal gradient of the mean LWP values in the area of the coastline in the vicinity of the radiometer location during the WH season: the larger LWP over land and the lower LWP over water surface.

5) During the CD season, the analysis of the AVHRR data in the vicinity of the radiometer location revealed an abnormal

LWP land-sea gradient and unexpected high LWP values over water surface. This effect stands in contrast to the results obtained by the SEVIRI instrument.

6) In order to find out the reasons for the abnormal land-sea LWP gradient in the vicinity of the radiometer location demonstrated by the AVHRR results, the LWP maps for the large terrain were analysed. Attention was paid to four water areas: the Neva bay, parts of the Ladoga Lake, Gulf of Finland, and Saimaa Lake. Abnormal land-sea LWP gradients

were detected for the ice-covered Neva bay and the Saimaa Lake. This phenomenon was attributed to the artefacts caused by the problems with the ice/snow mask used by the AVHRR retrieval algorithm.

7) The majority of the results of the study have been obtained when the HATPRO measurements were synchronized with the SEVIRI observations first: average HATPRO-SEVIRI time mismatch was less than 2 min while the HATPRO-AVHRR time mismatch was 9 min. It has been shown that time mismatch in the range 2-9 minutes did not affect the

results of the data comparison.

8) An attempt was made to evaluate qualitatively the influence of the cloud field inhomogeneity on the agreement between the satellite and the ground-based data. In order to detect the effect, the simple estimate of the LWP temporal variability was proposed as a measure of a cloud filed inhomogeneity. This estimate was based on the LWP values obtained by the ground-based radiometer and averaged over different time intervals. It has been found out that for both considered

satellite instruments the results are equally sensitive to the inhomogeneity of a cloud field. This conclusion is to a certain degree surprising since the SEVIRI measurements have lower spatial resolution than the AVHRR measurements, so the results of LWP retrieval by SEVIRI were expected to be more influenced by the inhomogeneity of a cloud field.

     As a final conclusion, we can assert that the LWP measurements by both satellite instruments SEVIRI and AVHRR agree well with the ground based observations by the microwave radiometer RPG-HATPRO during all seasons. The

AVHRR results have some preference if the correlations with ground-based measurements are compared but the SEVIRI observations have the smaller bias. Besides, the AVHRR LWP data of the version considered in the present study may have problems in winter over ice-covered water surfaces.



**Author contributions**

VSK and AK conceived the study together. VSK prepared the original draft of the manuscript with contributions from AK, MS and DVI. AK was in charge of the SEVIRI data analysis, MS was in charge of the AVHRR data analysis. Together VSK, AK, MS, and DVI interpreted the results, reviewed and edited the manuscript.

**Competing interests**

The authors declare that they have no conflict of interest.

**Acknowledgements**

The operation of the RPG-HATPRO instrument was provided by the Research Centre GEOMODEL of St. Petersburg State University (http://geomodel.spbu.ru/). The weather reviews were provided by the Russian North-West Administration on Hydrometeorology and Environmental Monitoring (http://www.meteo.nw.ru/).

**Funding**

The research was supported by Russian Foundation for Basic Research through the project No. 19-05-00372.

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





**Table 1.** Characteristics of the data agreement: correlation coefficient $r_c$, bias $b$ (satellite data minus ground-based data and SEVIRI data minus AVHRR data), and RMS difference $s$ obtained for the WH season (bias-corrected value $s_0$ is given in brackets).

| Compared data sets | $r_c$ | $b$, kg m$^{-2}$ | $s$, kg m$^{-2}$ |
|---|---|---|---|
| SEVIRI - HAT$_5$ | $0.45 \pm 0.10$ | -0.002 | 0.045  (0.045) |
| SEVIRI - HAT$_{10}$ | $0.55 \pm 0.09$ | -0.001 | 0.037  (0.037) |
| SEVIRI - HAT$_{20}$ | $0.63 \pm 0.08$ | -0.001 | 0.033  (0.033) |
| SEVIRI - HAT$_{60}$ | $0.66 \pm 0.07$ | -0.003 | 0.031  (0.031) |
| AVHRR - HAT$_5$ | $0.85 \pm 0.04$ | 0.014 | 0.036  (0.034) |
| AVHRR - HAT$_{10}$ | $0.91 \pm 0.02$ | 0.014 | 0.035  (0.032) |
| AVHRR - HAT$_{20}$ | $0.93 \pm 0.02$ | 0.015 | 0.037  (0.033) |
| AVHRR - HAT$_{60}$ | $0.92 \pm 0.02$ | 0.013 | 0.036  (0.034) |
| SEVIRI - AVHRR | $0.66 \pm 0.07$ | -0.016 | 0.049  (0.047) |

**Table 2.** The same as Table 1 but for the CD season.

| Compared data sets | $r_c$ | $b$, kg m$^{-2}$ | $s$, kg m$^{-2}$ |
|---|---|---|---|
| SEVIRI - HAT$_5$ | $0.70 \pm 0.07$ | 0.003 | 0.044  (0.044) |
| SEVIRI - HAT$_{10}$ | $0.70 \pm 0.07$ | 0.003 | 0.044  (0.044) |
| SEVIRI - HAT$_{20}$ | $0.70 \pm 0.07$ | 0.003 | 0.045  (0.044) |
| SEVIRI - HAT$_{60}$ | $0.69 \pm 0.07$ | 0.002 | 0.044  (0.044) |
| AVHRR - HAT$_5$ | $0.88 \pm 0.03$ | 0.017 | 0.058  (0.055) |
| AVHRR - HAT$_{10}$ | $0.88 \pm 0.03$ | 0.017 | 0.058  (0.055) |
| AVHRR - HAT$_{20}$ | $0.85 \pm 0.04$ | 0.017 | 0.060  (0.058) |
| AVHRR - HAT$_{60}$ | $0.84 \pm 0.04$ | 0.016 | 0.059  (0.057) |
| SEVIRI - AVHRR | $0.63 \pm 0.08$ | -0.014 | 0.070  (0.068) |

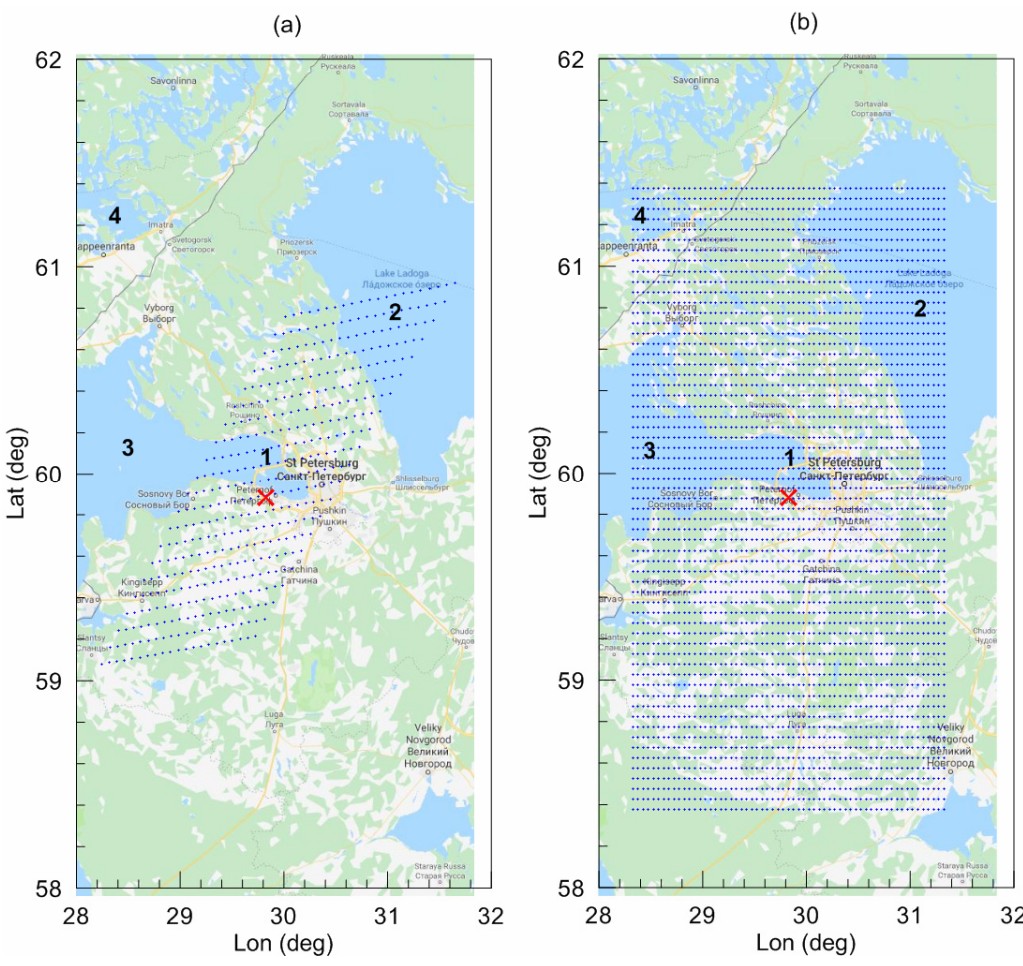

**Figure 1: The location of 441 SEVIRI measurement pixels (a) and 3721 AVHRR measurement pixels (b) selected for analysis in the large terrain. The position of the HATPRO radiometer is marked by the red cross. The black numbers identify the following objects: 1 – the Neva bay, 2 – the Ladoga Lake, 3 – the Gulf of Finland, 4 – the Saimaa Lake. Map data ©2019 Google.**



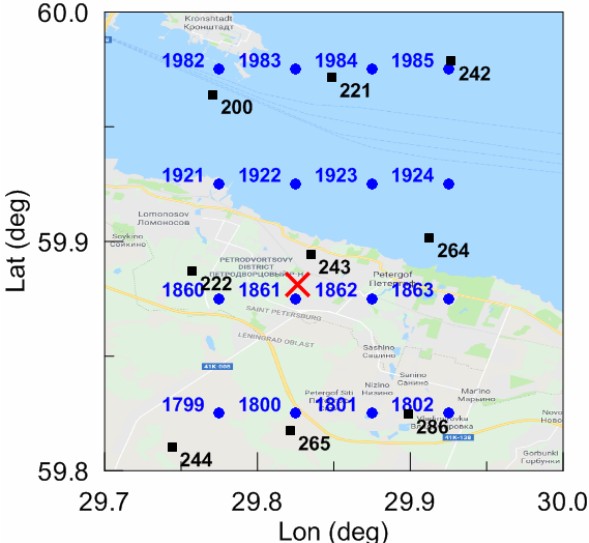

**Figure 2:** The location and numbers of 9 SEVIRI (black squares) and 12 AVHRR (blue circles) measurement pixels closest to the position of the HATPRO radiometer (marked by the red cross). The small terrain is shown. Map data ©2019 Google.

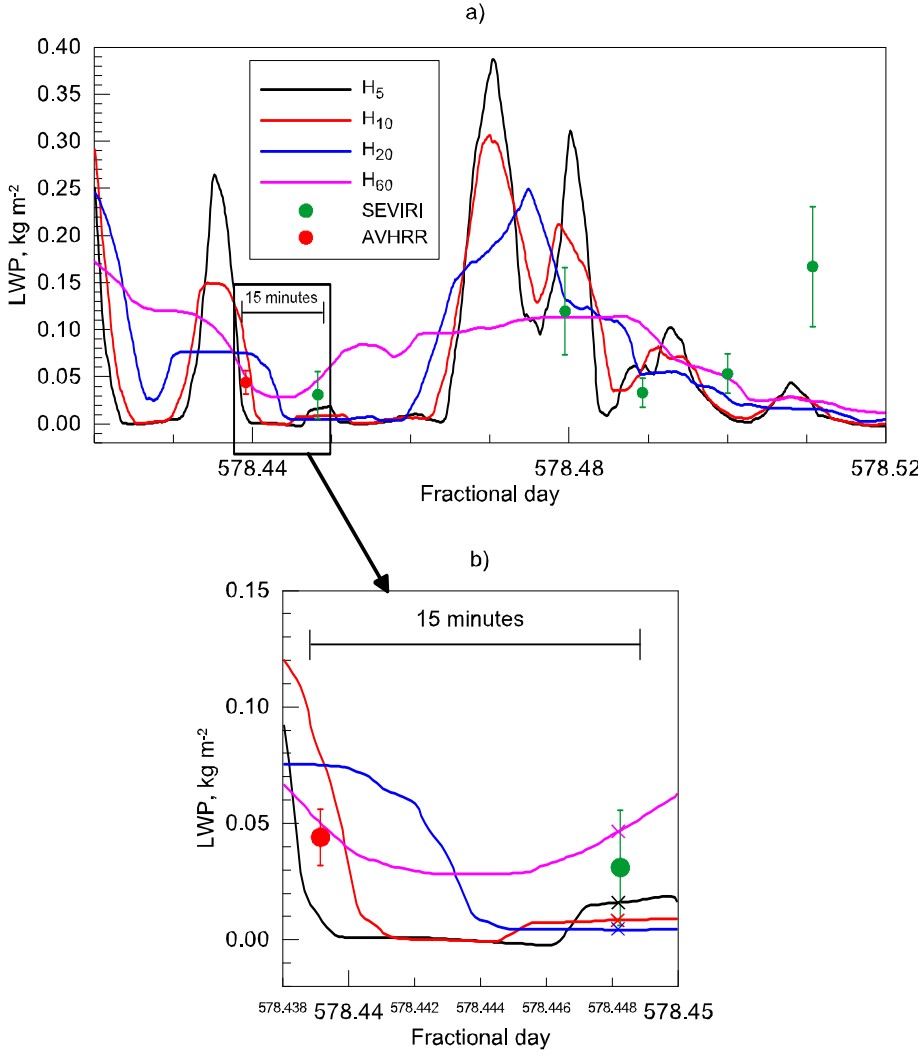

**Figure 3: The HATPRO data flow presented in the form of running average values with different averaging intervals (colour lines, see the legend). The SEVIRI and the AVHRR instantaneous measurements are shown as green and red dots. The 2.5 hour interval of observations on 2 July 2014 is displayed in panel "a". The 18 minute interval containing collocated SEVIRI and AVHRR measurements is magnified and is shown in panel "b". The colour crosses designate the averaged HATPRO measurements which are selected for comparison with the satellite data.**

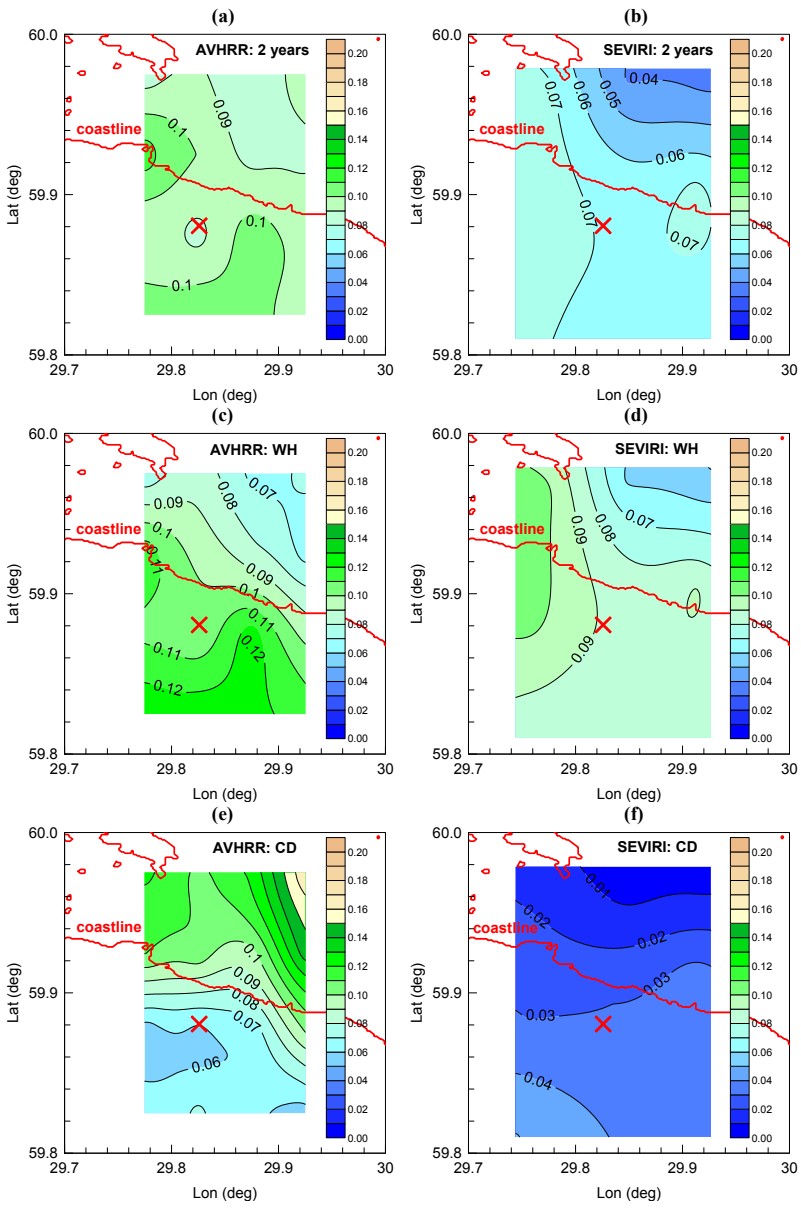

**Figure 4: The map of the mean LWP values (kg m$^{-2}$, colour scale) calculated for the small terrain and for the complete 2-year data set (a,b), the WH season (c,d) and the CD season (e,f): measurements by the AVHRR instrument (a,c,e) and the SEVIRI instrument (b,d,f). The position of the HATPRO radiometer is marked by the red cross, the coastline is marked by the red line. Vector shoreline data: (GSHHG, 2017).**

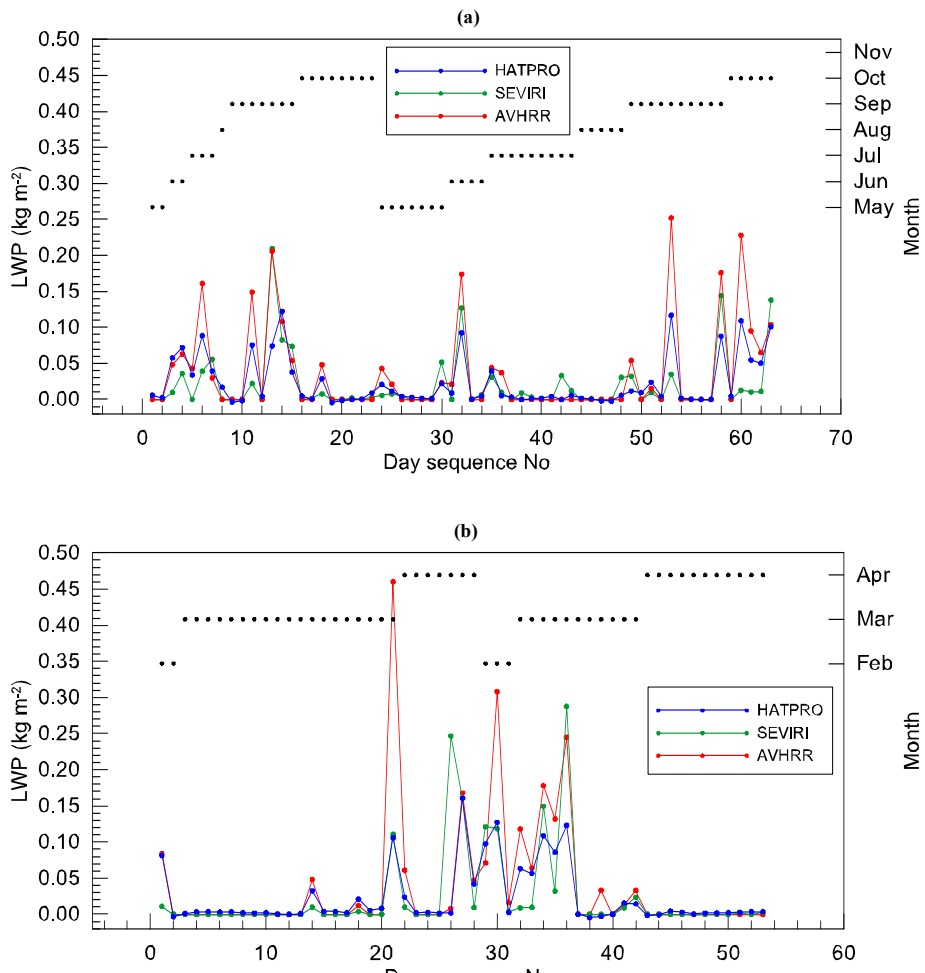

**Figure 5: The LWP values obtained by HATPRO (HAT$_{60}$, blue dots), SEVIRI (green dots) and AVHRR (red dots) as a function of day sequence number for the WH and CD seasons (a and b correspondingly). Colour dots are connected by lines only for demonstrative purpose. Black dots in combination with the right *y* axis indicate the month of measurements.**



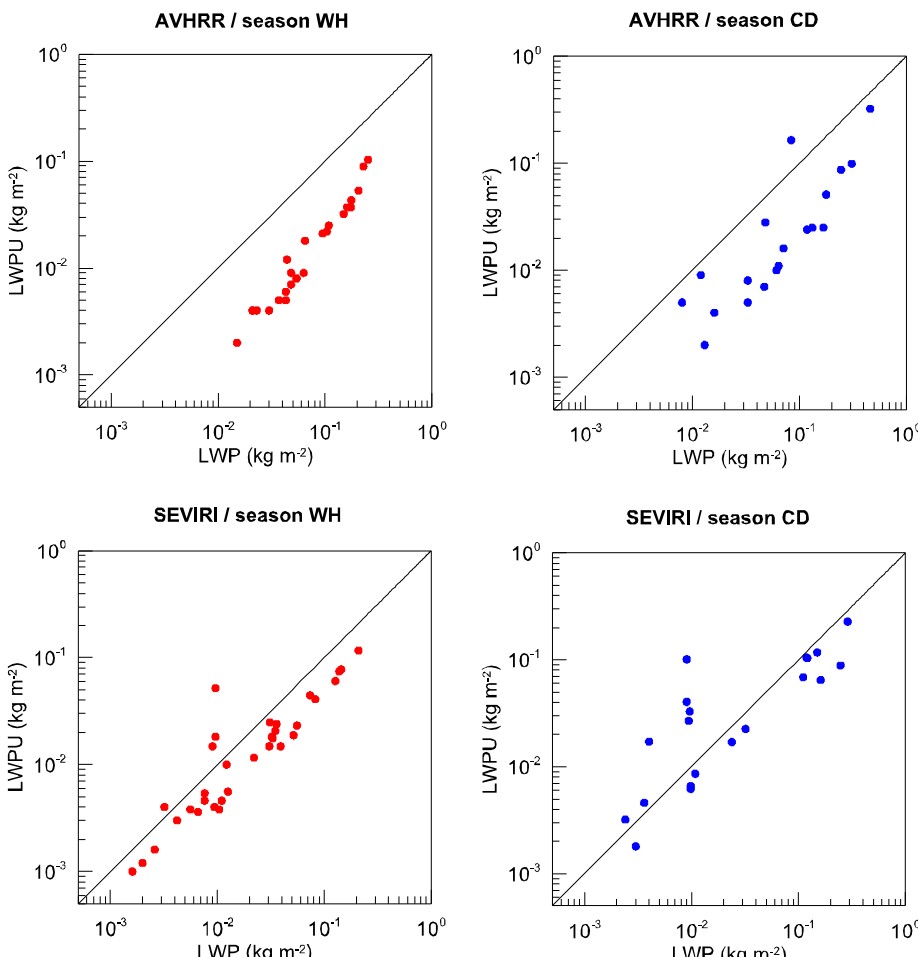

**Figure 6: The LWP retrieval uncertainty (LWPU) as a function of LWP value for the AVHRR and SEVIRI instruments and for different seasons (cloudy conditions only).**





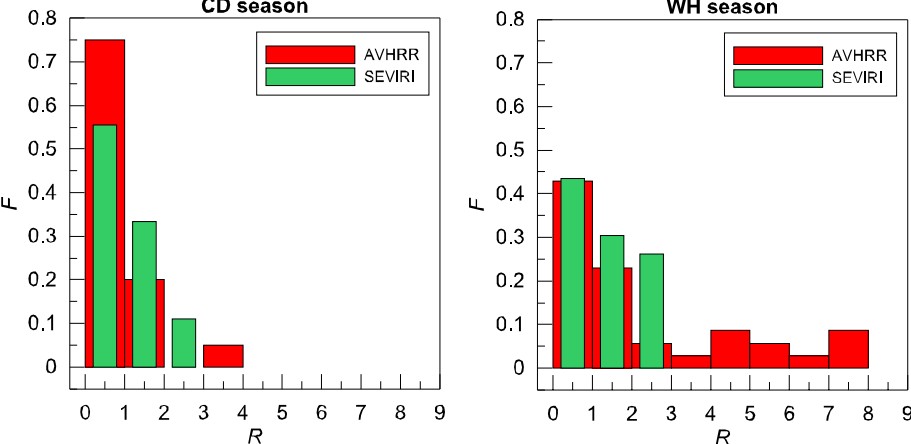

**Figure 7:** The relative frequency of occurrence (*F*) of the ratio (*R*) of the absolute difference between the satellite and ground-based data to the LWP uncertainty reported by the satellite instruments (cloudy conditions only).

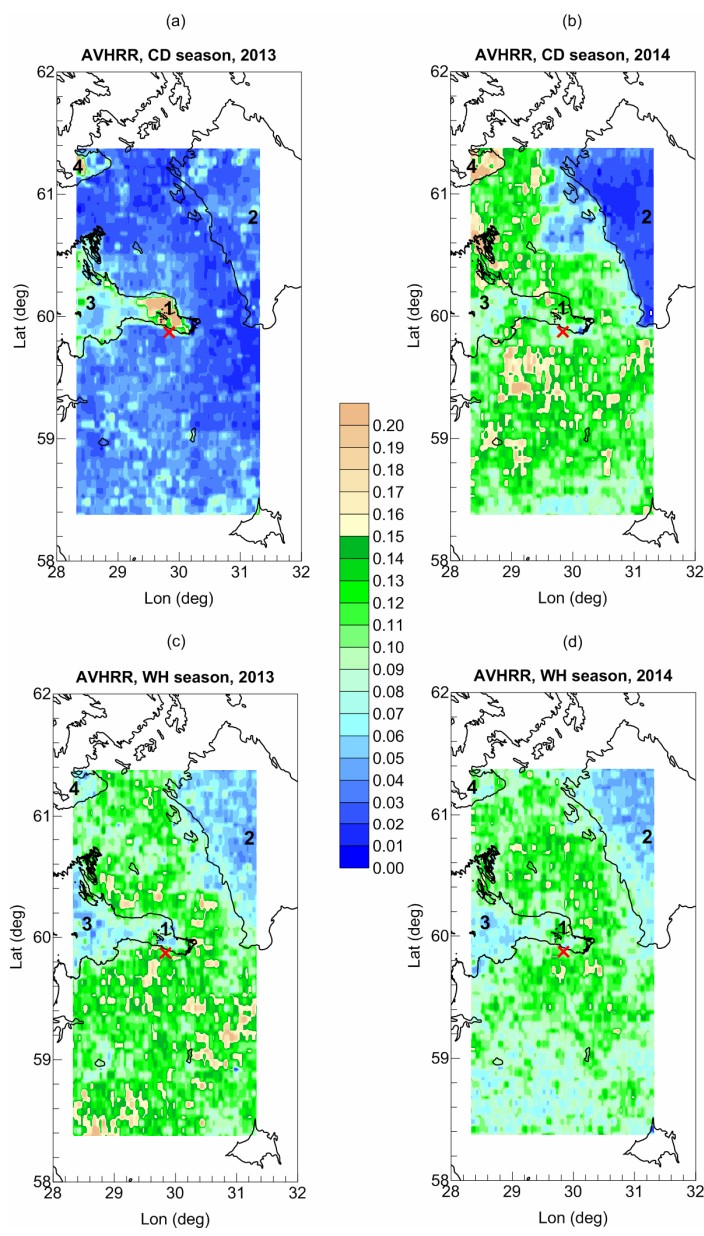

**Figure 8: The maps of the mean LWP values (AVHRR measurements, colour scale, kg m$^{-2}$) calculated for the large terrain for the CD and WH seasons of 2013 and 2014. The location of HATPRO radiometer is marked by the red cross. The black numbers identify the following objects: 1 – the Neva bay, 2 – the Ladoga Lake, 3 – the Gulf of Finland, 4 – the Saimaa Lake. Vector shoreline data: (GSHHG, 2017).**



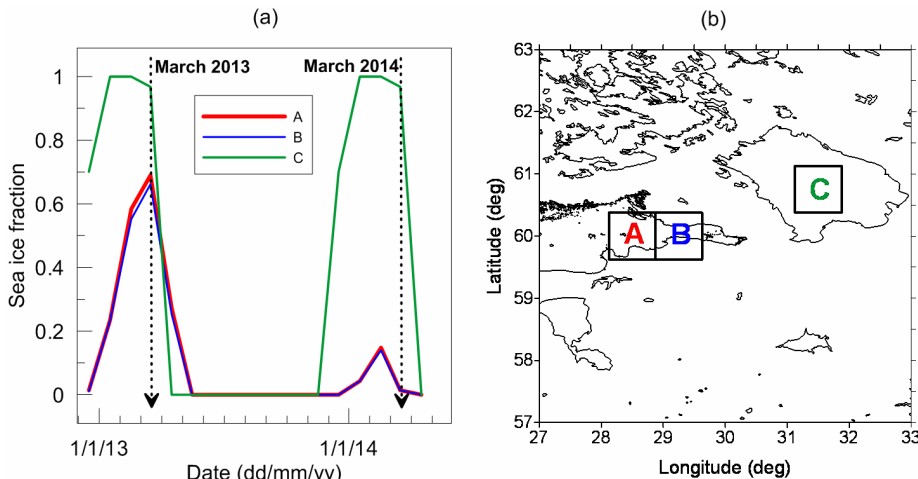

**Figure 9: (a) The monthly mean values of the "sea ice area fraction" taken from the ECMWF reanalysis for three pixels close to the radiometer site and plotted for the period of observations considered in the present study. (b) Location of the three pixels, colours of numbers correspond to colours of lines in panel "a". Vector shoreline data: (GSHHG, 2017).**





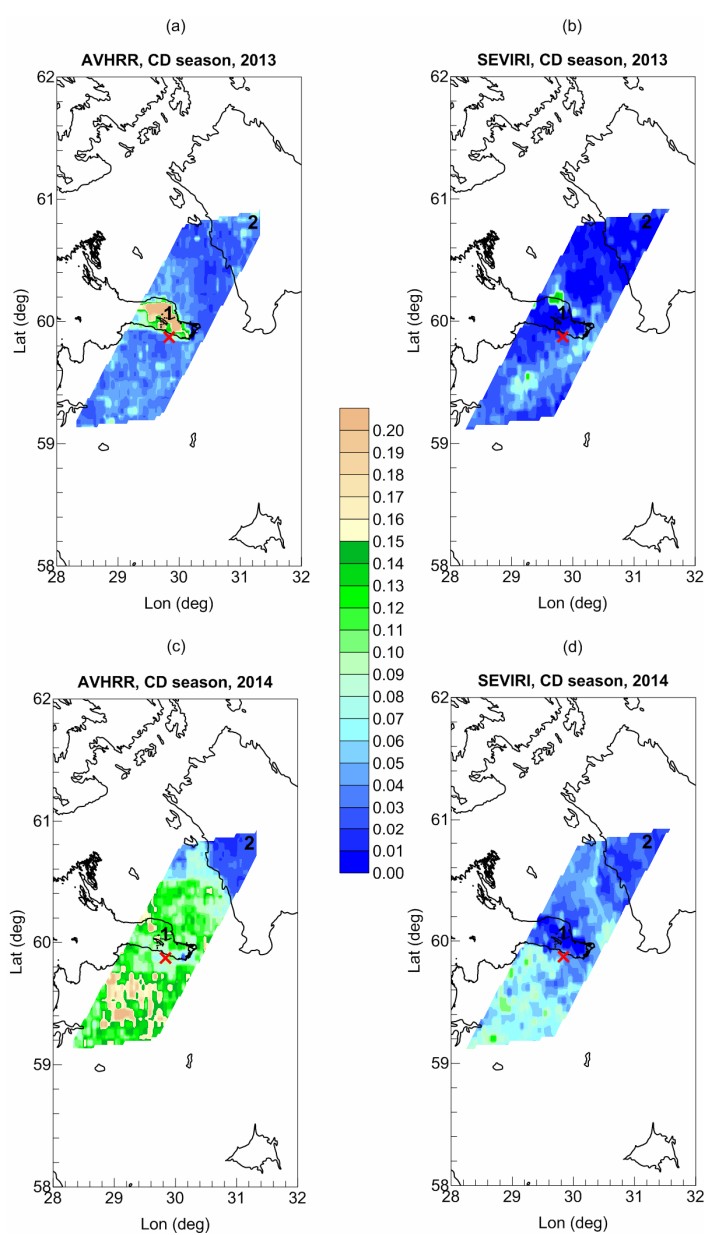

Figure 10: The maps of the mean LWP values (AVHRR and SEVIRI measurements, colour scale, kg m$^{-2}$) calculated for the large terrain for the CD season of 2013 (a, b) and 2014 (c, d). The location of HATPRO radiometer is marked by the red cross. The black numbers identify the following objects: 1 – the Neva bay, 2 – the Ladoga Lake. Vector shoreline data: (GSHHG, 2017).

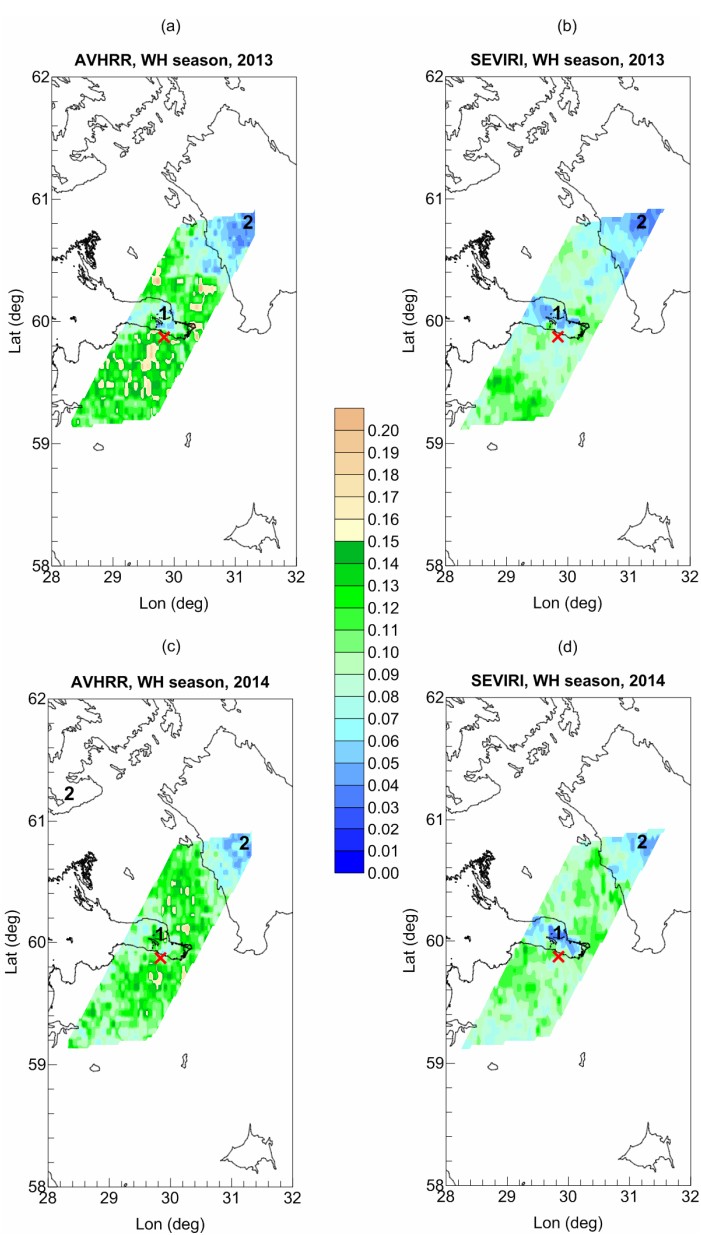

**Figure 11: The maps of the mean LWP values (AVHRR and SEVIRI measurements, colour scale, kg m$^{-2}$) calculated for the large terrain for the WH season of 2013 (a, b) and 2014 (c, d). The location of HATPRO radiometer is marked by the red cross. The black numbers identify the following objects: 1 – the Neva bay, 2 – the Ladoga Lake. Vector shoreline data: (GSHHG, 2017).**



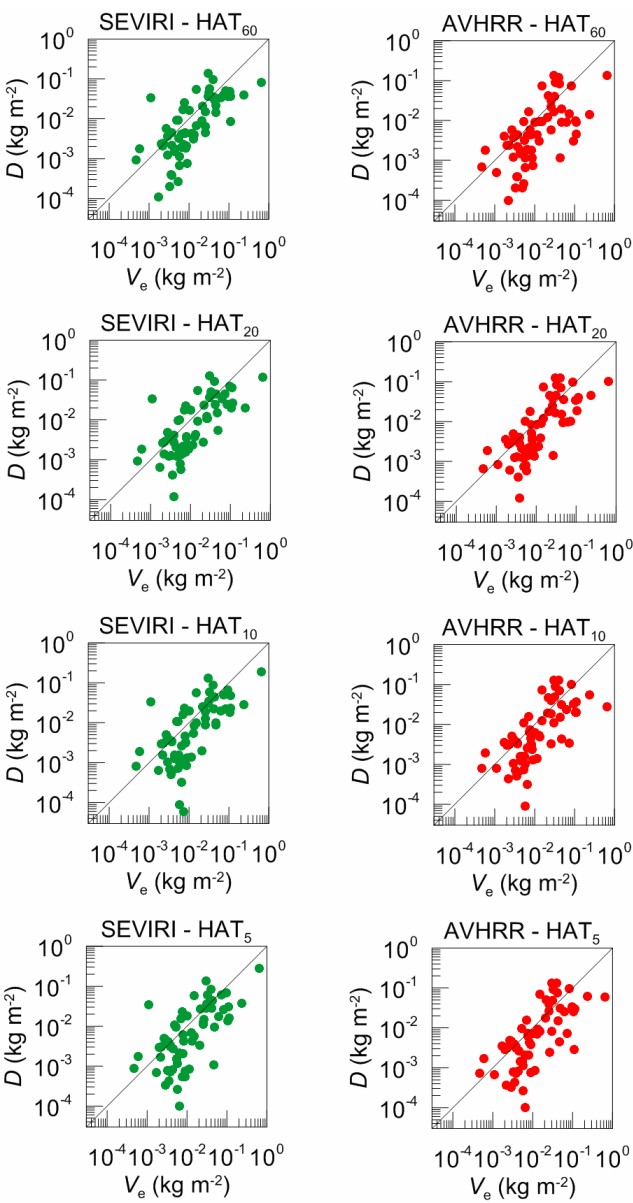

**Figure 12: The absolute difference $D$ between the ground-based and the satellite measurements of LWP as a function of the value of LWP variability estimate $V_e$ (see text). The data refer to the WH season.**