# Peer review of "Cross-comparison of cloud liquid water path derived from observations by two space-borne and one ground-based instrument in Northern Europe"

_Atmospheric Measurement Techniques, 2019_

## Referee Comment (RC1) · Anonymous Referee #3 · 5 Aug 2019

Cross-comparison of cloud liquid water path derived from observations by two space-borne and one ground-based instrument in Northern Europe

Kostsov et al., 2019

Major Scientific Contribution

This manuscript compares cloud liquid water path (LWP) from two satellite-based instruments, SEVIRI and AVHRR with measurements by a ground-based radiometer, RPG-HATPRO, located in St. Petersburg, Russia. The study is concerned with two issues: 1.) the different spatial resolutions of the satellite- and ground-based instruments and, 2.) the land-sea LWP gradient. A large terrain and a small one surrounding St. Petersburg were selected for the study. The ground-based data from RPG-HATPRO were averaged on 5-, 10-, 20- and 60-minute intervals (in order to find the optimal interval for best agreement), and the data were separated for cold and dry (CD) season and warm and humid (WH) season. It is found that the bias of the SEVIRI data relative the ground-based data is practically zero, while the AVHRR data show appreciable difference from the ground-based data, especially during the CD season, which is attributed by the authors to the coarse resolution of land-sea and snow/ice mask used by the AVHRR algorithm. It is also found that SEVIRI and AVHRR data are equally sensitive to the cloud field inhomogeneity.

Technical Comments

Page 9: Could you call the s in Eq. (2) just "RMS", and s0 in Eq. (3) "standard deviation"?

Page 16, Line 483: "filed" should be field?

Page 19: Table 1. Could you also add the mean of LWP from RPG-HATPRO data and the number of data points (N)?

Past tense is used in some places. Use present tense if possible.

———————————————————

---

## Referee Comment (RC2) · Anonymous Referee #1 · 11 Aug 2019

The authors compared the cloud liquid water path derived from SEVIRI and AVHRR with ground-based Hatpro measurements. The analyses are focused on the scale difference problem and the land-sea LWP gradient. The authors found that compared to Hatpro LWP, the SEVIRI LWP has no bias, AVHRR LWP has a positive bias. The The paper is well-written. A description of the SEVIRI dataset is missing, although the authors referred to their earlier paper.

Line 21: '. . . and a high bias () of the AVHRR results. . .'. high bias or positive bias?

Section 2: Which SEVIRI LWP dataset is used in this paper? Please give a short description of the SEVIRI LWP dataset in Section 2.

Lines 141-142, 3) The subset contains . . .: What is the size of the regular grid? '4) The AVHRR data are based on AVHRR GAC . . .' Move this item after '1) The data version is . . .'

Close to Line 275: In the WH season, $r_c$ seems increasing with averaging interval, but not in the CD season. There are different number of SEVIRI LWP data in WH and CD. Will this impact the $r_c$ in the WH and CD seasons?

Line 301-304 , explaining the features in Fig. 6: Why the LWP and LWPU have a linear relationship in a logarithmic scale?

Paragraph started from line 361: The authors suggest the land-sea LWP gradient is caused by the coarse resolution of the snow/ice mask used in AVHRR. Please explain what snow/ice cover data are used in the AVHRR LWP product and the spatial resolution of this data.

Paragraph started from line 383: The AVHRR images in Figs. 10 and 11 are similar to Fig. 8 but having less pixels. If you would like to save one figure, you could remove the AVHRR images and combine the SEVIRI images in one figure.

Line 433, '. . . between logarithms of D and Ve': D is not defined in the text.

Fig. 12 shows clearly that D and Ve are correlated. Perhaps it is also useful to make a linear fit of Ve and D, gives the slope and offset. How do we understand the correlation between D and Ve? As explained by the authors, there are other reasons cause the differences except for the cloud inhomogeneity. It might be possible to see the impact of cloud inhomogeneity if the LWP data are derived using the same algorithm and satellite data observed at different pixel size.

---

## Author Comment (AC1) · 4 Oct 2019

**The reply to the anonymous referee #3 (RC1)**

We are thankful to the referee for the comments. We appreciate all the comments; we took them into account while preparing the revised version of the manuscript.

Below, the actual comments of the referee are given in **bold** courier font and blue colour. The text added to the revised version of the manuscript is marked by red colour.

Major Scientific Contribution This manuscript compares cloud liquid water path (LWP) from two satellite-based instruments, SEVIRI and AVHRR with measurements by a ground-based radiometer, RPG-HATPRO, located in St. Petersburg, Russia. The study is concerned with two issues: 1.) the different spatial resolutions of the satellite- and ground-based instruments and, 2.) the land-sea LWP gradient. A large terrain and a small one surrounding St.Petersburg were selected for the study. The ground-based data from RPG-HATPRO were averaged on 5-, 10-, 20- and 60-minute intervals (in order to find the optimal interval for best agreement), and the data were separated for cold and dry (CD) season and warm and humid (WH) season. It is found that the bias of the SEVIRI data relative the ground-based data is practically zero, while the AVHRR data show appreciable difference from the ground-based data, especially during the CD season, which is attributed by the authors to the coarse resolution of land-sea and snow/ice mask used by the AVHRR algorithm. It is also found that SEVIRI and AVHRR data are equally sensitive to the cloud field inhomogeneity.

We appreciate that the referee highlighted the major points and results of our study.

Page 9: Could you call the s in Eq. (2) just "RMS", and s0 in Eq. (3) "standard deviation"?

We agree with this comment and changed the text before Eq. (2) and also the Table 1 caption accordingly:

- "The number of synchronised HATPRO-SEVIRI-AVHRR measurements was 63 during the WH season, and 53 during the CD season. The main statistical characteristics relevant to the agreement of the data are given in Tables 1 and 2. The bias b, the RMS s and the standard deviation s0 were calculated as follows:"
- "Table 1. Characteristics of the data agreement: correlation coefficient  $r_c$ , bias b (satellite data minus groundbased data and SEVIRI data minus AVHRR data), and RMS s obtained for the WH season (standard deviation  $s_0$  is given in brackets)."

We also made the appropriate change of terms throughout the entire text.

Page 16, Line 483: "filed" should be field?

Corrected.

Page 19: Table 1. Could you also add the mean of LWP from RPG-HATPRO data and the number of data points (N)?

The number of data points has been indicated just in the beginning of Section 4: "The number of synchronised HATPRO-SEVIRI-AVHRR measurements was 63 during the WH season, and 53 during the CD season." Following the advice of the referee, we indicated the number of data points also in table captions and added the mean LWP from HATPRO data:

- **"Table 1.** Characteristics of the data agreement: correlation coefficient  $r_c$ , bias *b* (satellite data minus ground-based data and SEVIRI data minus AVHRR data), and RMS *s* obtained for the WH season (standard deviation  $s_0$  is given in brackets). Total number of data points N is 63, the mean LWP values for HATPRO data sets are in the range 0.021-0.023 kg m-2."
- **"Table 2.** The same as Table 1 but for the CD season. Total number of data points N is 53, the mean LWP values for HATPRO data sets are in the range 0.022-0.023 kg m-2."

**Past tense is used in some places. Use present tense if possible.**

We have changed past tense to present tense in several places and we hope that this issue can be also addressed with the aid of a copy editor if the article is accepted for publication.

---

## Author Comment (AC2) · 4 Oct 2019

**The reply to the anonymous referee #1 (RC2)**

We are thankful to the referee for the careful reading of our manuscript and for the valuable comments. We appreciate all the comments; we took them into account while preparing the revised version of the manuscript.

Below, the actual comments of the referee are given in **`bold courier font and blue colour`**. The text added to the revised version of the manuscript is marked by red colour.

> **`The authors compared the cloud liquid water path derived from SEVIRI and AVHRR with ground-based Hatpro measurements. The analyses are focused on the scale difference problem and the land-sea LWP gradient. The authors found that compared to Hatpro LWP, the SEVIRI LWP has no bias, AVHRR LWP has a positive bias. The The paper is well-written. A description of the SEVIRI dataset is missing, although the authors referred to their earlier paper.`**

We are grateful to the referee for the positive assessment of our study. The comment about the description of the SEVIRI dataset is discussed below.

> **`Line 21: '…and a high bias () of the AVHRR results…'. high bias or positive bias?`**

"High" was substituted by "large positive".

> **`Section 2: Which SEVIRI LWP dataset is used in this paper? Please give a short description of the SEVIRI LWP dataset in Section 2.`**

We added a short description of the SEVIRI dataset just in the beginning of Section 2. For consistency, we also included a brief description of the HATPRO dataset. The text in the beginning of Section 2 now has two more items in the list of important points and reads as follows:

"The detailed description of the RPG-HATPRO and the SEVIRI datasets that are used in the present study has been presented in the article by Kostsov et al. (2018). Here we briefly enumerate the most important points:

1) The 14-channel microwave radiometer RPG-HATPRO (generation 3) has been routinely functioning at the measurement site of St.Petersburg State University (59.88°N, 29.83°E) since June 2012 with a sampling interval about 1-2 s and an integration time 1 s. The LWP values together with temperature and humidity profiles are derived from the microwave radiation brightness temperature measurements (zenith viewing mode) by the retrieval algorithm which is based on the inversion of the radiative transfer equation and uses the well known and widely applied approach of simultaneous retrieval of profiles of several atmospheric parameters that influence the radiative transfer at frequencies corresponding to spectral channels of a radiometer.

2) The SEVIRI-derived LWP measurements are part of the climate data record CLAAS 2 (CLoud property dAtAset using SEVIRI – Edition 2). It was created by the Satellite Application Facility on Climate Monitoring (CM SAF) based on the SEVIRI measurements on the geostationary MSG satellites. CLAAS data record was created from measurements of all SEVIRI instruments onboard the MSG 1-3

satellites and covers the time-span 2004 – 2015. SEVIRI scans the earth with a temporal resolution of 15 minutes. In the vicinity of St.Petersburg the ground pixel size is about 7 km. In the study by Kostsov et al. (2018), non-averaged LWP and CPH (cloud phase) fields (level 2 data) from the CLAAS 2 dataset were used for the time period of ground-based original data.

3) The time interval 1 December 2012 – 30 November 2014 was taken for the analysis. …

4) …"

**Lines 141-142, 3) The subset contains…: What is the size of the regular grid? '4) The AVHRR data are based on AVHRR GAC…' Move this item after '1) The data version is…'**

Item (4) has been placed after item (1). The size of the regular grid (0.05°) has been indicated:

"4) The subset contains sampled data on a regular grid with a step of 0.05° (no averaging done). In each grid cell, the AVHRR pixel with the smallest satellite zenith angle is collected."

**Close to Line 275: In the WH season, r_c seems increasing with averaging interval, but not in the CD season. There are different number of SEVIRI LWP data in WH and CD. Will this impact the r_c in the WH and CD seasons?**

To our opinion, the difference between numbers of data points for the WH and CD seasons is small and produces no impact on the correlation coefficients. We have made the appropriate note in the text in Section 4 close to line 280 of the revised text:

"To our opinion, the difference between total numbers of data points for the WH and CD seasons is small and is not the reason for the obtained seasonal features."

**Line 301-304 , explaining the features in Fig. 6: Why the LWP and LWPU have a linear relationship in a logarithmic scale?**

Addressing this point we added the following sentence after "First, the dependence of LWPU on LWP on a logarithmic scale is very close to linear for both instruments.":

"Since the analysis of the LWP retrieval algorithms used for processing satellite data is beyond the scope of our study, we can not discuss the reasons for such dependence."

**Paragraph started from line 361: The authors suggest the land-sea LWP gradient is caused by the coarse resolution of the snow/ice mask used in AVHRR. Please explain what snow/ice cover data are used in the AVHRR LWP product and the spatial resolution of this data.**

In this comment the esteemed referee pointed at the absence of any confirmation of our suggestion about the coarse spatial resolution of the AVHRR snow/ice data. We absolutely agree with this comment. However, we do not have at our disposal the snow/ice data which have been used in the AVHRR LWP retrieval process. These data do not belong to the level 2 products. That is why in the submitted manuscript we declared: "*The discussion of the problem of identification of clouds and ice/snow covered surfaces is beyond the scope of our study.*" Nevertheless, taking into account the importance of the problem and the important comment made by the referee, we added some general information about the procedure of creating cloud/snow/ice masks. The new version of the paragraph "*The discussion of the problem of identification of clouds and ice/snow covered surfaces...*" is the following:

"The analysis and discussion of the problem of identification of clouds and ice/snow covered surfaces is beyond the scope of our study. We only note that it is an important problem which attracted much attention in the studies relevant to remote sensing of atmospheric state and composition from satellites. A very detailed overview on existing algorithms for cloud and snow detection on AVHRR imagery can be found in the paper by Musial et al. (2014). Based on this paper, we outline several principal features relevant to the considered problem:

1) A cloud mask allows discrimination between surface and cloud signals and it is a common input to the generation of satellite products.

2) A misclassification in a cloud/snow mask propagates to higher-level products and may alter their usability.

3) The majority of algorithms for cloud and snow detection incorporate a series of spectral, textural and/or temporal tests which are arranged in a decision-tree scheme.

4) Ancillary data are required for a threshold parameterisation, which might be divided into meteorological and surface data sets. An instantaneous atmospheric state can be estimated either by climate models or by rough approximations based on climatological mean values. Usually, such simulations are of low spatial resolution. Another source of inaccuracies is temporal sampling of a climate model.

We do not have ice/snow data used in the AVHRR retrieval algorithm at our disposal, but item (4) in the presented list is a kind of confirmation of our suggestion about the coarse resolution of the ice/snow data being the reason for abnormal LWP land-sea gradients over two relatively small water bodies."

**Paragraph started from line 383: The AVHRR images in Figs. 10 and 11 are similar to Fig. 8 but having less pixels. If you would like to save one figure, you could remove the AVHRR images and combine the SEVIRI images in one figure.**

The referee is perfectly right, the AVHRR images are similar. We deliberately organised Figs. in such a way that visual comparison is more convenient. So we prefer to leave Figs. "as is".

**Line 433, '… between logarithms of D and Ve': D is not defined in the text.**

We agree with this comment and edited the text which now reads:

"We analysed both the WH and CD seasons and obtained similar results, therefore only the results corresponding to the WH season are demonstrated and discussed. Each data point in Fig. 12 shows the LWP variability estimate $V_e$ at the moment of a measurement and the correspondent absolute difference $D$ between the satellite measurement of LWP and the ground-based measurement averaged over 5, 10, 20 or 60 min defined as follows:

$$D = \left| LWP_{sat} - HAT_j \right|$$

(7)

where *sat* refers to SEVIRI or AVHRR, *j* indicate the averaging interval as in Eq. (5). Obviously, one can not expect…"

**Fig. 12 shows clearly that D and Ve are correlated. Perhaps it is also useful to make a linear fit of Ve and D, gives the slope and offset. How do we understand the correlation between D and Ve? As explained by the authors, there are other reasons cause the differences except for the cloud inhomogeneity. It might be possible to see the impact of cloud**

inhomogeneity if the LWP data are derived using the same algorithm and satellite data observed at different pixel size.

We have followed the advice of the referee and in the new Fig. 12 we have plotted the linear fit for the logarithms of $D$ and $V_e$. (power fit for $D$ and $V_e$). The values of the coefficients are given in the text:

"The parameters of the linear fit $\ln(D)=B\ln(V_e)+A$ are similar for all data sets. The values of $B$ are in the range 0.76 – 0.87 and the values of $A$ are in the range (-1.9) – (-1.2)."

In order to address the question of the referee about understanding of the correlation between $D$ and $V_e$, we added the following text which also includes the idea for detection of the impact of cloud inhomogeneity given by the referee:

"This correlation is an indication of the noticeable influence of the inhomogeneity of the cloud field on the difference between the ground-based and the satellite data. We estimate this influence as "noticeable" since it is not masked by other error sources which are obviously present, as it was already mentioned above. It might be possible to see the pure impact of cloud inhomogeneity if the two satellite experiments are completely identical except pixel size, however this is not our case."